# A Longitudinal Examination of Perceived Parent Behavior and Positive Youth Development: Child-Driven Effects

**Goda Kaniušonytė [1,*], Mary Page Leggett-James [2] and Brett Laursen [2]**

1 Institute of Psychology, Mykolas Romeris University, LT-08303 Vilnius, Lithuania
2 Department of Psychology, Florida Atlantic University, Boca Raton, FL 33314, USA; marypagejames@gmail.com (M.P.L.-J.); laursen@fau.edu (B.L.)
* Correspondence: godakan@mruni.eu

**Abstract:** The advent of the 21st Century brought a new interest in promoting Positive Youth Development and a renewed emphasis on understanding transactional relations between parenting and adolescent development. The present study examined conventional parent-driven pathways, which describe the putative role of parents in the formation of positive characteristics in children, as well as the prospect of child-driven effects, which describe how parents respond to evidence of Positive Youth Development by potentially increasing support and reducing psychological control. We tested these pathways in a sample of 458 Lithuanian adolescents (52.2% girls; M = 15.14 years old at the outset) who completed surveys assaying perceptions of parent behaviors and self-reports of positive development (character, competence, connection, caring, and confidence) at annual intervals from ages 15–18. Across most lags, children's perceptions of parenting changed in response to their own positive development with increased support and decreased psychological control. In contrast, there were no longitudinal associations from perceptions of parenting to subsequent Positive Youth Development. The results offer insight into parenting in the 21st Century, a time when youth are increasingly encouraged/required to acquire volunteer experiences designed to promote positive development. To the extent that these experiences are successful, one unexpected offshoot may be better relationships with parents.

**Keywords:** adolescents; Positive Youth Development; parental support; psychological control; bidirectional effects

## 1. Introduction

In the 21st Century, youth are increasingly viewed not as problems to be solved, but as resources to be fostered (Lerner et al. 2005). This shift and the dynamic developmental systems-based ideas that undergird it afford an optimistic view about the ways that parents can promote Positive Youth Development in ways that constructively contribute to the neighborhood, community, and society. Conventional wisdom holds that parents shape the development of positive attributes in their adolescent children. Parent-driven effects have a long and influential history in the literature on parenting and parenting styles (Power 2013). The field of Positive Youth Development is no exception; many studies endorse the view that parents are an important source of influence over the acquisition of adaptive attributes (e.g., Bebiroglu et al. 2013; Bowers et al. 2014). Yet, for much of this time, scholars have warned of the dangers of focusing solely on parents as socialization agents, arguing that just as children react to parents, many parent behaviors are also reactive, arising in response to child attributes and behaviors (e.g., Pettit and Arsiwalla 2008). Child-driven models hold that children influence parents—in their own right—resulting in a transactional, cross-lagged, longitudinal feedback loop wherein children influence and are influenced by parents. Transactional models are ascendant in 21st Century scholarship on parenting and adolescent development (Overton 2015), but they are not well represented in empirical studies that assess Positive Youth Development as originally defined by the Five Cs of

character, competence, connection, caring, and confidence (Lerner et al. 2005). The present study examines mutually influential associations between adolescent positive development and perceptions of parenting behaviors across four consecutive years spanning the end of secondary school.

### 1.1. 21st Century Models of Adolescent Development: A Focus on the Positive

A major shift in our understanding of optimal adolescent development occurred early in the 21st Century, as scholars sought alternatives to models that emphasized the acquisition of problem behaviors. Mid- and late-20th Century models defined optimal developmental outcomes as the absence of undesirable behaviors, prompting policy makers and practitioners to create programs to reduce the frequency of adjustment problems such as mental health challenges, suicide, teenage pregnancy, substance and alcohol abuse, and delinquency. Although laudatory in their goals and outcomes, this view inadvertently characterized adolescence as a period full of problems to be fixed, instead of opportunities waiting to be fulfilled. Paradoxically, an emphasis on adolescence as a period of normative disturbance may have inadvertently created a self-fulfilling prophecy for some (Lerner et al. 2006). The term "Positive Youth Development" emerged in response, to emphasize the promise of youth and to capture the opportunities available to those in this age period (Lerner et al. 2005).

Positive Youth Development is a strengths-based model built on the assumption that all adolescents possess the potential for healthy, successful development (Lerner 2021). It is defined through the psychological, behavioral, and social characteristics known as the Five Cs (competence, confidence, connection, character, and caring). The model recognizes the importance of connectivity between the individual and the environment, also known as person ⇔ context coactions, the most important of which is captured in close, interpersonal relationships (Lerner et al. 2015). The main premise of the Five Cs of the Positive Youth Development model is that youth will thrive when their strengths align with key resources, such as positive and sustained relations with caring adults, life-skill-building experiences, and opportunities to participate and take leadership in family, school, and community activities (Lerner 2021).

Fast-forwarding to the 21st Century, increasingly, youth are viewed in terms of their potential for positive development. Every adolescent has a unique set of strengths that can be harnessed for the benefit of themselves, their interpersonal relationships, and their community (e.g., Bornstein 2003; Flanagan and Faison 2001). Youth with specific interests and talents are encouraged to channel them in ways that constructively afford skill development and encourage engagement with family members and the community. Children and adolescents who participate in school clubs and other structured activities report greater involvement in community groups and closer parent–child relationships, compared with those who are not similarly engaged (Moore and Glei 1995). Thus, adolescent participation in structured Positive Youth Development activities yields benefits for the community and the family. Parents are assumed to play an important role in successful youth development by fostering a sense of belongingness and meaningfulness and by promoting the development of self-regulation skills connected to competence (Lewin-Bizan et al. 2010). Positive youth characteristics, in turn, are believed to promote successful interpersonal relationships, including those between parents and children.

Volunteerism is a prominent example of the emphasis on promoting positive development. In recent years, there has been a push for children and adolescents to be involved in community activities. Accompanying the shift away from programs focused on reducing negative behaviors, practitioners and policy makers have turned to civic engagement as a means of promoting Positive Youth Development (Lerner et al. 2003). These commitments stem from the belief that competence, confidence, connection, character, and caring all flow from community engagement. Secondary education continues to highlight the importance of volunteerism because high-quality volunteering opportunities foster civic engagement in high school students, which promotes positive attitudes among youth and later civic

engagement in adulthood (Gallant et al. 2010; Lerner 2004). Much less attention has been given to understanding how parenting and parent–child relations can similarly promote Positive Youth Development.

*1.2. 21st Century Models of Parent–Child Relationships: A Focus on Transactions*

The outset of the 21st Century also marked a renewed emphasis on conceptual models that emphasize mutual parent–child influence. To be sure, transactional models have long been discussed in the field (Bell 1968; Sameroff 1975). However, applications to relationships during adolescence are a recent development (Laursen and Collins 2009). Transactional models posit a longitudinal, mutually reciprocal interplay between parent and child behavior, characterized by bidirectional influence processes (Sameroff and Mackenzie 2003). The model does not start with one partner or the other, but recognizes that parents act on and react to child behavior and that children act on and react to parent behavior, which produce reciprocal influence pathways. It is important to note that although transactions are often depicted in terms of the same variables (e.g., negativity on the part of one partner elicits negativity from the other), reciprocal interconnections can and do exist between different variables. Herein, we operationalize transactional processes in terms of longitudinal cross-lagged effects, recognizing that other scholars have other strategies for representing these processes.

One notable transactional model that emphasizes Positive Youth Development is the dynamic relational developmental systems metatheory, which holds that development is a reflection of interpersonal contexts and the interactions that take place within them (Overton 2015). In this system-based perspective, development is conceived of as a dynamic process, wherein the individual is in a constant state of becoming (moving "from potential to actuality"). Multiple interpersonal contexts shape this developmental process, key among them during the first two decades of life being relationships with parents. The process is neither static, nor unidirectional. The individual alters the developmental context, but the context places important constraints on patterns of development; together, they form a bidirectional, dynamic system capable of optimizing the realization of individual potential.

We focused on two forms of constructive parenting (as reported by adolescents). Expressions of support encompasses nurturing behaviors that convey emotional warmth and psychological acceptance and reassuring behaviors that encourage individuation and autonomous action (Barber et al. 2005). Support is assumed to bolster self-worth and achievement while protecting against adjustment difficulties. Constructive parents support autonomy development by offering choices, providing explanations for requests, and validating feelings and views. Supportive parenting also promotes self-regulation, with concomitant benefits to psychosocial adjustment (e.g., Lewin-Bizan et al. 2010; Steinberg et al. 1989), which are presumably a product of the provision of informational and instrumental resources, communication, and emotional validation.

The avoidance of psychological control entails respect for and refraining from behaviors that intrude on the child's psychological world (Soenens and Vansteenkiste 2010). Constructive parents avoid guilt induction and love withdrawal, behaviors designed to constrain, invalidate, and emotionally manipulate the child to feel, think, and behave as the parent wishes (Barber 1996). The avoidance of psychological control is particularly important during the adolescent years because parents who refrain from discouraging child initiatives are implicitly granting opportunities to make independent decisions, thus fostering a sense of autonomy (Hare et al. 2014). The presence of psychological control is known to be associated with a host of adjustment difficulties (e.g., Kaniušonytė and Laursen 2021; Pettit et al. 2001). On the other side, adolescents with parents who refrain from psychological control have better decision-making skills and higher self-esteem than those with controlling parents (Luyckx et al. 2007; Silk et al. 2003), presumably because there are no family-imposed psychological barriers to the optimal realization of potential in these domains.

Scholars differ as to the relative advantages of parent and child reports of parenting. The use of child report measures may inflate shared variance with self-reports of adjustment symptoms, although findings suggest that there is enough overlap in perceptions of overt behaviors that the resulting associations are not unduly influenced by shared reporter variance (Valdes et al. 2016). Moreover, parent reports of family interactions are not especially accurate; child reports have greater convergence with observer reports than do those of parents (Gonzales et al. 1996). Finally, when it comes to understanding the mechanisms whereby parent behaviors drive child outcomes, it may well be the case that child perceptions are better indicators of child outcomes than are parent perceptions because it is the child's interpretation of events and behaviors that dictate adjustment outcomes (Stattin and Kerr 2000). Therefore, in the present study, we used child reports of parenting, mindful of the limitations described above.

### 1.3. Research on Transactional Models of Parent–Adolescent Relationships and Positive Youth Development

Although several longitudinal studies have documented longitudinal, cross-lagged transactional associations between parenting and adolescent behavior problems (e.g., Gorostiaga et al. 2019; Huey et al. 2020), to our knowledge, there are no comparable studies of Positive Youth Development operationalized in terms of the Five Cs. Below, we summarize the literature, starting with a brief overview of concurrent research on correlated associations between parenting and Positive Youth Development components (Five Cs), followed by longitudinal research that focuses exclusively on parent-driven effects. No longitudinal studies could be identified describing child-driven effects of Positive Youth Development on parenting behaviors.

Concurrent correlational studies describe associations between parenting and Positive Youth Development. Parenting style (Kiadarbandsari et al. 2016), perceived parental school involvement, and perceived parental monitoring and warmth are associated with Positive Youth Development (Bowers et al. 2014). Longitudinal studies report similar parent-driven effects. Longitudinal studies have found that perceived parental warmth and monitoring have been tied to increases in global Positive Youth Development (a composite of competence, confidence, connection, character, and caring) from ninth to eleventh grade (Napolitano et al. 2011). In younger adolescents, perceived parent psychological control and behavioral control were indirectly linked to later Positive Youth Development (Cao et al. 2020; Lewin-Bizan et al. 2010)

### 1.4. The Current Study

The present study tested a transactional model, informed by relational developmental systems, which encompassed both child-driven and parent-driven cross-lagged effects. To be specific, we hypothesized that perceived positive parenting practices foster subsequent adaptive youth behaviors and that Positive Youth Development elicits subsequent perception of constructive parent behaviors, operationalized as bidirectional parallel processes (see Figure 1). The model tests the assumption that adolescent children influence and are influenced by perceived parenting behaviors. Although many longitudinal studies have explored similar transactional processes in the context of problem behavior (e.g., Huey et al. 2020; Soenens et al. 2008), our study is unique in its focus on longitudinal, cross-lagged, transactional developmental processes that describe Positive Youth Development specifically operationalized in terms of the original (Lerner et al. 2005) Five Cs. Based on past research, we anticipated transactional pathways between perceived parenting practices and Positive Youth Development, although we recognize that unidirectional studies tend to overstate the magnitude of effects because the effects were inflated by correlated patterns of change (Hafen and Laursen 2009). Our focus on the family context offers an important complement to the existing emphasis and empirical body of knowledge on Positive Youth Development and engagement with the community outside of the home (e.g., Ramey and Rose-Krasnor 2012).

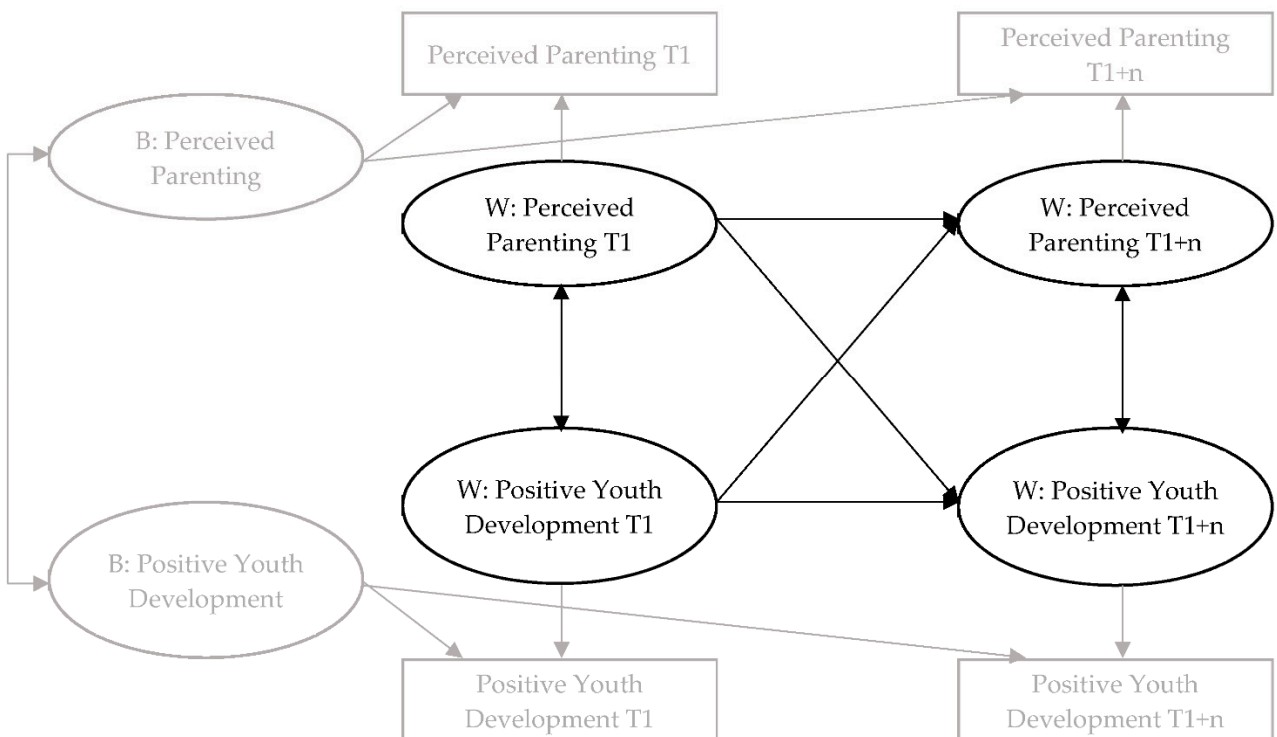

**Figure 1.** The conceptual and measurement model of mutually influential relationships. Note: The conceptual model is represented by black lines and the measurement model by black and grey together. The measurement model is the random intercept cross-lagged panel model. W = Within-person. B = Between-person.

The study was conducted in Lithuania, a Northern European country that is currently a member of NATO and the European Union. Most of the parents described in the current study were raised when the country was part of the Soviet Union, a time when conformity and obedience were prioritized (Gorlizki and Khlevniuk 2020). Psychological control was a salient construct in the Soviet Union (Hart et al. 1998), and although three decades of freedom have brought numerous essential changes to life in postcommunist societies such as the establishment of democracy, individualism, and the adoption of other Western values, most Eastern European countries still report somewhat higher levels of "traditional" parenting practices, compared to their Western European counterparts (Steinbach and Maslauskaitė 2020). Nevertheless, adolescent development in Lithuania resembles that in other Western communities where youth attend school in relatively small, relatively homogeneous cities (Kaniušonytė and Žukauskienė 2018). Of relevance to our study, a 2014 review of youth development programs in Lithuania did not identify any that focused on Positive Youth Development (Gabrialavičiūtė et al. 2014). In the intervening years, only one such program was developed (Truskauskaitė-Kunevičienė et al. 2020).

## 2. Materials and Methods

### 2.1. Participants

The community sample comprised 454 (215 boys and 239 girls) students attending five high schools in Western Lithuania. Participants ($M_{age}$ = 15.14, $SD_{age}$ = 0.48) were in the 9th grade (i.e., the 1st year of high school) at the outset. Nearly all participants were ethnic Lithuanian; 69.5% of the participants lived with two biological parents; 26.1% of children received free nutrition at school.

### 2.2. Procedure

Consistent with national and regional policy, parents were informed about the study by letter and asked to contact the school or the investigators if they did not want their child

to participate. Written assent was received from students, who were told that participation was voluntary. Trained research assistants administered questionnaires in class during regular school hours in the spring of 2013, 2014, 2015, and 2016.

A total of 454 students participated in the 9th grade (participation rate = 99.1%). Of this total, 446 participated as 10th graders, 391 participated as 11th graders, and 371 participated as 12th graders. Thus, 81.7% of the original sample participated in 4 waves of data collection. There were no differences in any study or demographic variables between students with and without complete data at all time points, except that students missing at least one wave of data were more likely to receive free nutrition at school (d = 0.54) than those with complete data.

An average of 17.1 % of reports were missing on perceived parenting variables (range = 5.6–29.4%), and an average of 14 % were missing on Positive Youth Development (range = 4.8–23.2%). To further explore the patterns of missingness and determine whether the data were missing at random, we conducted a normed $\chi^2$ ($\chi^2/\text{df}$ ratio) test; there is no firm consensus on the recommended values required, but there is agreement that a value less than 2.0 indicates that data are missing at random and that maximum likelihood techniques are appropriate for use (Bollen 1989). The normed $\chi^2$ value was 0.83, so missing item-level data were imputed with an EM algorithm and missing wave-level data were handled with Full-Information Maximum-Likelihood estimation (FIML). As recommended by Enders (2010), we included variables with nonsystematic missingness in the models to meet the requirements for missing data applications under missing-at-random conditions.

### 2.3. Measures

Positive Youth Development was measured at each wave using the Measure of Positive Youth Development (Lerner et al. 2005; Phelps et al. 2009). The scale consists of 78 items that cover five aspects of Positive Youth Development: character (20 items), competence (11 items), connection (22 items), caring (9 items), and confidence (16 items). The bifactor structure suggested by Geldhof et al. (2014) and validated with this sample by Erentaitė and Raižienė (2015) was modeled. In a bifactor model, the global construct of Positive Youth Development is modeled as a direct function of items rather than only being modeled as a function of lower-order latent constructs (Five C's). Thus, each item indicates a lower-order construct and a general construct by loading onto each simultaneously. Longitudinal and gender measurement invariance can be found in Supplementary Materials S1 and S2. For this study, factor scores of the global Positive Youth Development from the strong longitudinal invariance model were used for all subsequent analyses. Internal reliability was acceptable (alpha = 0.74–0.75).

Parent psychological control was measured at each wave with the 8-item Psychological Control Scale-Youth Self-Report (Barber 1996), describing emotional control and guilt induction (e.g., "Always trying to change how I feel or think about things") for mothers and fathers separately. Items were rated on a scale ranging from 1 (not like her/him) to 3 (a lot like her/him). Scores for mothers and fathers were combined at the item level, and measurement invariance was tested using combined scores. Longitudinal and gender measurement invariance can be found in Supplementary Materials S1 and S2. The internal reliability was acceptable (alpha = 0.79–0.90).

Parent support was measured at the second, third, and fourth waves using the 16-item Transformational Parenting Questionnaire (Morton et al. 2011), describing behaviors that indicate affection, caring, and encouragement (e.g., "Shows comfort and understanding when I am upset") for mothers and fathers separately. Items were rated on a scale ranging from 1 (strongly disagree) to 5 (strongly agree). Scores for mothers and fathers were combined at the item level, and measurement invariance was tested using combined scores. Longitudinal and gender measurement invariance can be found in Supplementary Materials S1 and S2. The internal reliability was acceptable (alpha = 0.74–0.75).

### 2.4. Plan of Analysis

The analyses were conducted with Mplus Version 8.4 (Muthén and Muthén 1998–2017) using Robust Maximum Likelihood (MLR) estimation within a structural equation model framework. The model fit was examined by using the Comparative Fit Index (CFI), and the Root-Mean-Squared Error of Approximation (RMSEA). CFI values higher than 0.90 are indicative of an acceptable fit with values higher than 0.95 suggesting an excellent or very good fit. RMSEA values lower than 0.05 indicate a good or close fit, and values as high as 0.08 represent acceptable fit. In addition, we examined the 90% confidence interval of the RMSEA: the model fit can be considered acceptable when the upper bound of this confidence interval is no greater than 0.1 (Kline 2016). As a convention, we report the chi-squared statistic; however, we did not use it to test the model fit since it is overly sensitive in moderately large samples (Chen 2007). To determine significant differences between models, at least two of the following criteria had to be matched: $\Delta\chi^2$ significant at $p < 0.05$ (Satorra and Bentler 1994), $\Delta$CFI $\geq$ 0.010, and $\Delta$RMSEA $\geq$ 0.015 (Chen 2007). If the models did not differ significantly, the more parsimonious model with more degrees of freedom was retained.

To investigate the within-person longitudinal associations between Positive Youth Development and perceptions of parenting, we conducted a Random Intercept Cross-Lagged Panel Model (RI-CLPM) for each parent behavior separately. RI-CLPM uses latent variables to distinguish stable between-person trait-like differences in constructs across waves from variation within a person at each wave on those same behaviors. Figure 1 depicts the measurement model. The autoregressive parameters represent the amount of within-person carry-over effect, and cross-lagged parameters indicate the extent to which variables predict one another within the same person over time. Correlations involving latent between-person variables describe whether adolescents who are higher overall on one construct (across waves and compared to other persons) are also higher (or lower) overall on another construct (Hamaker et al. 2015). In order to enhance model parsimony, we tested whether cross-lagged effects, autoregressive paths, and T2–T3 within-time correlations (correlated changes) were time invariant. Thus, we compared the baseline unconstrained model (M1) with the model assuming time invariance of cross-lagged associations (M2), T2–T3 within-time correlations (M3), autoregressive paths (M4), and all within-person paths together (M5).

## 3. Results

### 3.1. Preliminary Analyses

Concurrent bivariate correlations are presented in Table 1. At all times, Positive Youth Development was positively correlated with perceived parent support and negatively correlated with perceived psychological control.

### 3.2. Transactional Associations between Perceived Parent Psychological Control and Positive Youth Development

The model fit the data well ($\chi^2(9) = 19.9$, CFI = 0.996, RMSEA = 0.052). The findings supported the assumption of time invariance only for cross-lagged paths (Table 2). The estimates of cross-lagged effects, autoregressive paths, and within-time correlations for between and within-person effects are reported in Figure 2. Within-person results indicated that higher levels of Positive Youth Development (relative to the person's average levels) predicted decreased perceived psychological control (lower relative to the person's average than before). Positive Youth Development and perceived psychological control were negatively correlated at T2 and at T4 (relative to one's own average score). At the between-person level, adolescents with higher overall levels of Positive Youth Development (compared to other adolescents) perceived their parents as less psychologically controlling.

**Table 1.** Within and over time bivariate correlations between Positive Youth Development and perceived parental behavior.

| Variable | 1 | 2 | 2 | 4 | 5 | 6 | 7 | 8 | 9 | 10 |
|---|---|---|---|---|---|---|---|---|---|---|
| 1. PYD T1 | – | | | | | | | | | |
| 2. PYD T2 | 0.86 [0.83, 0.88] | – | | | | | | | | |
| 3. PYD T3 | 0.77 [0.73, 0.81] | 0.89 [0.87, 0.91] | – | | | | | | | |
| 4. PYD T4 | 0.68 [0.62, 0.73] | 0.72 [0.66, 0.77] | 0.78 [0.73, 0.82] | – | | | | | | |
| 5. Support T1 | 0.41 [0.32, 0.50] | 0.44 [0.35, 0.53] | 0.41 [0.32, 0.49] | 0.32 [0.24, 0.41] | – | | | | | |
| 6. Support T2 | 0.38 [0.29, 0.47] | 0.40 [0.31, 0.49] | 0.43 [0.35, 0.52] | 0.35 [0.26, 0.43] | 0.56 [0.48, 0.63] | – | | | | |
| 7. Support T3 | 0.27 [0.18, 0.36] | 0.28 [0.18, 0.37] | 0.33 [0.24, 0.42] | 0.50 [0.43, 0.57] | 0.34 [0.26, 0.44] | 0.41 [0.31, 0.50] | – | | | |
| 8. Control T1 | −0.34 [−0.44, −0.24] | −0.35 [−0.44, −0.26] | −0.33 [−0.42, −0.23] | −0.25 [−0.34, −0.15] | −0.36 [−0.45, −0.27] | −0.28 [−0.37, −0.18] | −0.20 [−0.29, −0.11] | – | | |
| 9. Control T2 | −0.33 [−0.41, −0.26] | −0.41 [−0.48, −0.33] | −0.38 [−0.45, −0.30] | −0.31 [−0.38, −0.24] | −0.41 [−0.51, −0.31] | −0.32 [−0.41, −0.23] | −0.23 [−0.34, −0.14] | 0.52 [0.42, 0.61] | – | |
| 10. Control T3 | −0.35 [−0.44, −0.26] | −0.42 [−0.50, −0.35] | −0.42 [−0.49, −0.34] | −0.33 [−0.42, −0.24] | −0.36 [−0.44, −0.27] | −0.39 [−0.48, −0.30] | −0.26 [−0.37, −0.17] | 0.58 [0.51, 0.66] | 0.59 [0.51, 0.67] | – |
| 11. Control T4 | −0.32 [−0.40, −0.23] | −0.34 [−0.42, −0.26] | −0.37 [−0.44, −0.28] | −0.44 [−0.52, −0.36] | −0.27 [−0.36, −0.18] | −0.29 [−0.38, −0.20] | −0.51 [−0.61, −0.40] | 0.39 [0.30, 0.48] | 0.47 [0.38, 0.55] | 0.56 [0.46, 0.64] |

Note: PYD—Positive Youth Development, Support = Perceived Parent Support, Control = Perceived Parent Psychological Control, T = time; all correlations significant at $p < 0.001$.

**Table 2.** Model fit of the random intercept cross-lagged panel models and model comparisons.

| Model | $\chi^2$ (*df*) | CFI | RMSEA [95% CI] | Model Comparison | $\Delta\chi^2$ | ΔCFI | ΔRMSEA |
|---|---|---|---|---|---|---|---|
| | | | *Positive Youth Development and Psychological Control* | | | | |
| Model 1 | 19.9 (9) | 0.996 | 0.052 [0.020–0.083] | | | | |
| Model 2 | 26.1 (13) | 0.995 | 0.047 [0.019–0.073] | M1/M2 | 60.17 | 0.001 | 0.005 |
| Model 3 | 40.9 (11) | 0.988 | 0.077 [0.053–0.103] | M1/M3 | 200.99 * | 0.008 | 0.025 |
| Model 4 | 39.2 (13) | 0.990 | 0.067 [0.043–0.091] | M1/M4 | 190.30 * | 0.006 | 0.015 |
| Model 5 | 69.6 (19) | 0.980 | 0.077 [0.058–0.096] | M1/M5 | 490.75 * | 0.016 | 0.025 |
| | | | *Positive Youth Development and Parent Support* | | | | |
| Model 1 | 1.2 (1) | 1 | 0.021 [0.000–0.128] | | | | |
| Model 2 | 2.0 (3) | 1 | 0.000 [0.000–0.068] | M1/M2 | 00.84 | 0 | 0.021 |
| Model 3 | 15.8 (2) | 0.992 | 0.123 [0.072–0.183] | M1/M3 | 140.63 * | 0.008 | 0.102 |
| Model 4 | 18.1 (3) | 0.991 | 0.105 [0.062–0.154] | M1/M4 | 160.93 * | 0.009 | 0.084 |
| Model 5 | 47.7 (6) | 0.975 | 0.124 [0.093–0.158] | M1/M5 | 460.55 * | 0.025 | 0.123 |

Note: N = 454, 95% confidence intervals given in brackets. M1 = baseline model; M2 = model with cross-lagged paths fixed to be time invariant; M3 = model with T3–T4 within-time correlations fixed to be time invariant; M4 = model with autoregressive paths fixed to be time invariant; M5 = model with cross-lagged paths and T2–T4 correlations fixed to be time invariant. * $p < 0.05$.

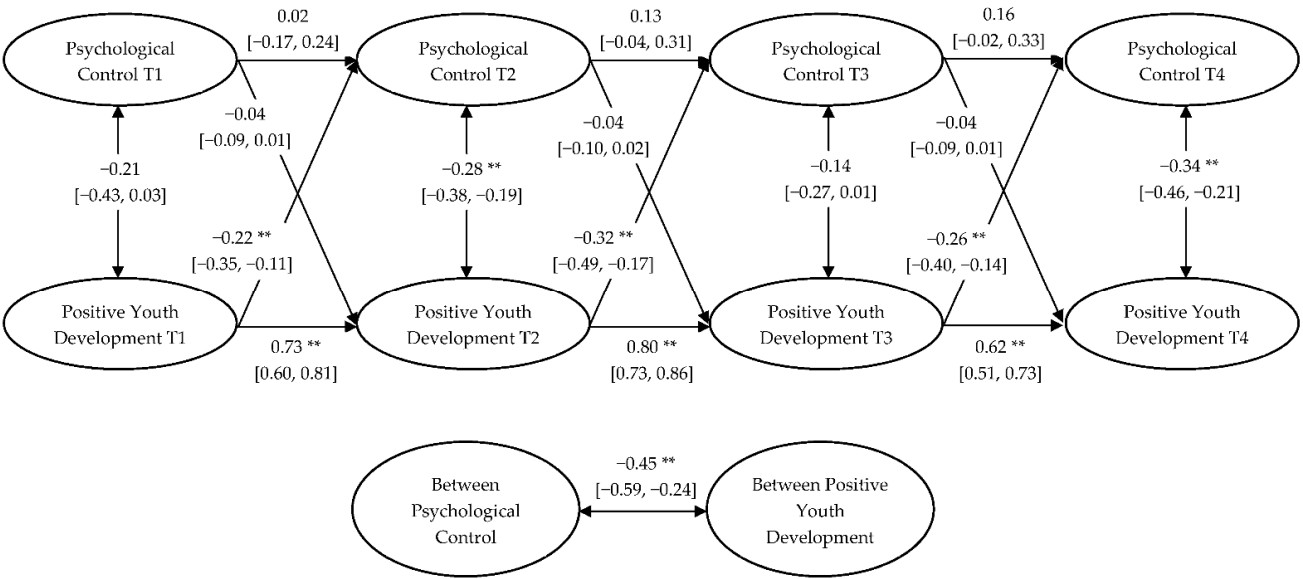

**Figure 2.** Simplified random intercept cross-lagged panel model between perceived psychological control and Positive Youth Development with standardized coefficients. Note: Cross-lagged paths are fixed to be equal between time points; the discrepancies are due to standardization. $N = 454$. ** $p < 0.001$.

### 3.3. Transactional Associations between Perceived Parent Support and Positive Youth Development

The model fit the data well ($\chi^2(1) = 1.2$, CFI = 1, RMSEA = 0.021). The findings supported the assumption of time invariance only for cross-lagged paths (Table 2). The estimates of cross-lagged effects, autoregressive paths, and within-time correlations for between and within-person effects are reported in Figure 3. Within-person results indicated that higher levels of Positive Youth Development (relative to the person's average levels) predicted increased perceived support (higher relative to the person's average than before). A change in Positive Youth Development and perceived support (increases or decreases relative to one's own average) were positively correlated at all times. At the between-person level, adolescents did not display general between-person level differences, meaning that adolescents perceived their parents as similarly supportive across different levels of Positive Youth Development.

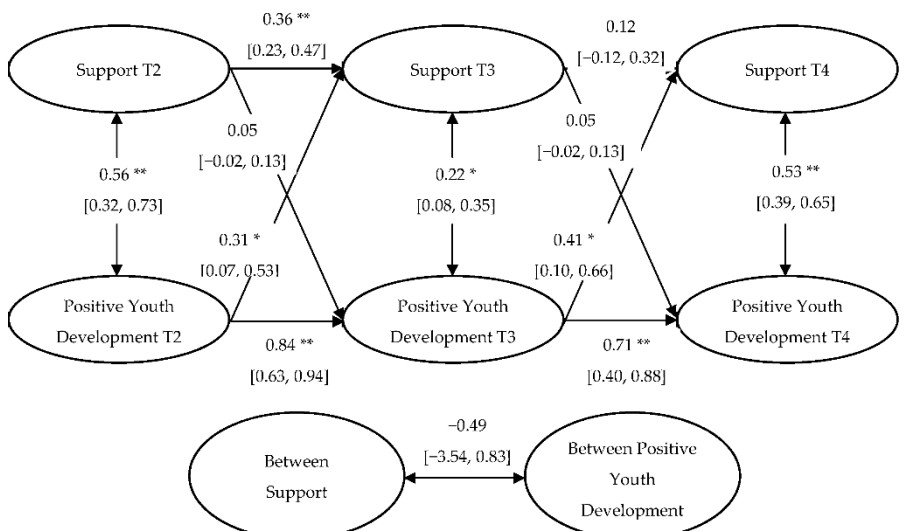

**Figure 3.** Simplified random intercept cross-lagged panel model between perceived support and Positive Youth Development with standardized coefficients. Note: Cross-lagged paths are fixed to be equal between time points, the discrepancies are due to standardization. $N = 454$. * $p < 0.05$, ** $p < 0.001$.



## 4. Discussion

The results of a four-wave cross-lagged transactional longitudinal analysis revealed child-driven, but not parent-driven effects: Positive Youth Development predicted subsequent changes in both perceived parent support and psychological control. Perceived parent behavior did not predict changes in Positive Youth Development. Our study is unique in that it is one of the first to examine transactional associations between Positive Youth Development and adolescent perceptions of constructive parenting practices. Nevertheless, the findings are consistent with narrative reviews (Meeus 2016) arguing that child-driven effects are more consistent than parent-driven effects in models exploring associations between adolescent difficulties and parent–child relationships.

The results afford at least two interpretations, which are not mutually exclusive. First, Positive Youth Development shapes child perceptions of parent behavior. In this sense, Positive Youth Development has potential to affect the overall quality of parent–child relationships. Thus, as more psychosocial resources become available to an individual, the better their relationships will be with all significant others, including parents (Kochendorfer and Kerns 2017). Better-quality relationships with parents translate into more positive perception of parents and, in some cases, more constructive parenting practices. Second, Positive Youth Development directly shapes specific forms of parent behavior. Although much of the literature focuses on the ways in which negative adolescent adjustment shapes parenting behavior, there is some evidence that positive adolescent adjustment works in a similar fashion (Yan and Ansari 2016). Well-adjusted adolescent behaviors elicit greater parental warmth and support, both of which are reflections of constructive parenting (Barber et al. 2005; Lewin-Bizan et al. 2010). It may be that when children and adolescents demonstrate positive characteristics, they elicit and reward the use of positive parenting techniques in their parents (Yan and Ansari 2016).

When adolescents are encouraged to create positive relationships with their family, friends, and community, there are demonstrated benefits for personal development (Lerner 2004). The "Big Three" features of effective programs for Positive Youth Development include opportunities for youth to participate in leadership activities, programs that emphasize life skills, and participation in sustained youth–adult relationships. Many volunteer activities embrace all three. These may have important spinoff effects. When parents are part of a support system of Positive Youth Development, they have new opportunities to practice positive parenting, avoiding the decline in warmth and support that can afflict poor-quality child relationships that struggle with negativity during adolescence (Laursen et al. 2010).

We did not find reciprocal transactions interactions between Positive Youth Development and parenting. Parenting practices were unrelated with changes in Positive Youth Development. The findings may illustrate decreases in the amount of time that adolescents spend in the company of parents and the increased exposure to and influence of others, especially friends and romantic partners). Alternatively, Meeus (2016) argued that parents play an important role in positive aspects of development during the late childhood and early adolescent years, prior to the emergence of a mature self-image, but that this influence declines with age. As autonomy and independence develop, plasticity declines and is less affected by external stimuli. That is not to say that parents have no influence on adolescents, but rather, adolescents may be less influenced by parent behaviors in late adolescence as compared to early adolescence. It is also possible that parents influence their adolescents in different ways than parents in previous times. Daily social media use is common during adolescence, an activity that did not exist for previous parenting generations. Parents who monitor their adolescent's social media content may engage youth with activities geared for Positive Youth Development, thus influencing their adolescent in unmeasured ways. A final alternative recognizes findings from a genetically informed study that suggested that parent-driven effects are illusory, a byproduct of error arising from gene–environment correlations (Guimond et al. 2016).

*Limitations, Future Directions, and Implications*

This study provides new insights into the interplay of family relationships and Positive Youth Development, and it should be considered both in light of its strengths and its shortcomings. Perceived parenting behaviors and adolescent outcomes were collected simultaneously.

Some may argue that our reliance on self-reports is a limitation. However, parents are not particularly accurate reporters of adolescent's inner states such as internalizing symptoms (Angold et al. 1987), and the impact of parenting depends more on how the adolescent perceives and interprets the parent behaviors than on how the parent reports their own behaviors (Stattin and Kerr 2000). To be sure, our results describe adolescent perceptions of parenting behaviors, which should not be confused with actual or observed parent behaviors. There is merit to understanding both. Of additional concern is bias arising from same reporter variance, which may inflate variance across predictor and outcome variables. There is evidence that bias in within-reporter correlations between mother and child views of psychological control and child behavior problems depend on the degree to which the latter is readily observable; shared views across reporters minimize the chances that within-reporter results are a product of same-reporter biases (Valdes et al. 2016). Finally, it is worth noting that the participants attended school in a small, homogeneous Northern European community. It remains to be seen whether the findings generalize to youth living in heterogeneous, urban contexts.

Mediators and moderators of Positive Youth Development and parental behavior should be included in future research. Potential mediators could include self-regulation (Bowers et al. 2011), identity-formation processes (Berzonsky et al. 2007; Luyckx et al. 2007), or the satisfaction of basic needs (Costa et al. 2016). The timing and units of change can be an emphasis of future research on this topic as well. It might be the case that perceived parental influence on adolescent's Positive Youth Development happens earlier than the ninth grade and the parent-driven effects might be evident at earlier age periods. Furthermore, it is possible that the transactions that happen within the relationships can be captured only in shorter time lags. Because interinfluence between two variables in dynamic relational developmental processes occurs continually over time the discrete time measurements can capture only a snapshot of it (Rioux and Little 2020).

The findings should not be interpreted to mean that contemporary parents are ineffectual or irrelevant to adolescent children. There are several domains where parents clearly foster adaptive competencies in their adolescent children, such as academic competence, connection, and self-regulation (see Laursen and Collins 2009, for a review). As noted above, it may also be the case that parents impact some, but not all of the competencies included in our global measure of positive development. Further, as we have suggested elsewhere (Lewin-Bizan et al. 2010), the overall tenor and quality of parent relationships with adolescent children has long-lasting repercussions for adolescent adjustment. Finally, we note that parents are expected to demonstrate diminished influence over adolescents because in many Western cultures, the goal of competent parenting is precisely the encouragement of this sort of disengagement and self-reliance. Put simply, positive parenting behaviors facilitate adolescent autonomy, which may ultimately reduce adolescent dependence on parents and parent influence over adolescent children (Lewin-Bizan et al. 2010).

The findings have important implications for 21st Century parents who face a landscape very different from that of parents in earlier generations. Adolescents today have access to technology that offers the opportunity for near-constant contact with peers and nonstop entertainment (Brown et al. 2013). Thus, parent influence is challenged not only by the rise in peer influence during adolescence, but also by influence from social media. In this context, parents potentially may more effectively serve as moderators of outside sources of influence, rather than as forces that directly shape outcomes (e.g., Dickson et al. 2015; Marion et al. 2014). Effective parents recognize that social media can be used as an effective tool to facilitate Positive Youth Development by engaging adolescents with volunteerism and community activities of their interests (Lee and Horsley 2017). Parents may direct

youth to community media sources that will ultimately promote Positive Youth Development. Thus, compared to previous generations, contemporary parents are faced with new challenges and new opportunities to indirectly shape the positive development of their adolescent children. These mechanisms can be an important question to raise in future empirical studies.

The findings offer two important takeaways. The first is that parents respond to emerging competence in children with adaptive adjustments to parenting practices, such as more support and less psychological control. Although it is good to know that parents are responsive to child development, we believe that all adolescents would profit from parent support and an environment free of psychological control. Practitioners should alert parents to their subtle adjustments in the face of child maturation and encourage an awareness of constructive parent behaviors regardless of the child's level of positive development. The second takeaway is that parents should embrace practices such as volunteerism that encourage Positive Youth Development because child competencies might have a constructive impact on perceived or actual parenting practices or at least the tenor of the parent–child relationship (Theokas and Lerner 2006), improving the chances that both parties will enjoy the time they share together.

**Supplementary Materials:** The following are available online at https://www.mdpi.com/article/10.3390/socsci10100369/s1, Supplementary Materials S1: Tests of Longitudinal Measurement Invariance for all Study Variables, Supplementary Materials S2: Tests of Gender Measurement Invariance for all Study Variables.

**Author Contributions:** Conceptualization, G.K. and B.L.; methodology, G.K.; formal analysis, G.K.; data curation, G.K.; writing—original draft preparation, G.K.; writing—review and editing, G.K., M.P.L.-J., and B.L. All authors have read and agreed to the published version of the manuscript.

**Funding:** Brett Laursen received support for the preparation of this article from the Lithuanian Research Council (09.3.3-LMT-K-712-05-0006).

**Institutional Review Board Statement:** The study was conducted according to the guidelines of the Declaration of Helsinki, and approved by the Ethics Committee of Mykolas Romeris University (protocol code SR-2822, 2012-10-02).

**Informed Consent Statement:** Informed consent was obtained from all subjects involved in the study.

**Data Availability Statement:** Data will be made available upon request.

**Conflicts of Interest:** The authors declare no conflict of interest.

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
