# Peer review of "A Longitudinal Examination of Perceived Parent Behavior and Positive Youth Development: Child-Driven Effects"

_socsci, doi:10.3390/socsci10100369_

Round 1

Reviewer 1 Report

I have no concerns regarding this manuscript. I'm thus happy to recommend acceptance as it is. This work represents a very good inclusion in the Special Issue!

I congrats the authors.

Author Response

We are very grateful for this positive evaluation.

Reviewer 2 Report

The study focuses on a novel and interesting aspect: the positive development of adolescents. It suggests that the participation of adolescents in volunteer tasks promotes their positive development and improves relationships with their parents.

Some aspects should be considered

31 many parent behaviors are reactive, arising in response to child attributes and behaviors: Is it worth noting that the children's behaviors are even more reactive to the parents' behavior?

55-57 Parents are assumed to play an important role in successful youth development by fostering a sense of belongingness and meaningfulness, and by promoting the development of self-regulation skills connected to competence  Due to the relevance of the aforementioned questions for the strong personality of the children, should it be noted that the benefits of volunteering should be considered a complement, an improvement, to that essential function of parents?

469  the goal of competent parenting is precisely the encouragement of this sort of disengagement and self-reliance It is not clear that these two characteristics are beneficial for adolescents

Author Response

We are grateful for the valuable insights. Our responses are below.

31 many parent behaviors are reactive, arising in response to child attributes and behaviors: Is it worth noting that the children's behaviors are even more reactive to the parents' behavior?

R. We revised the sentence to read as follows: Yet, for much of this time, scholars have warned of the dangers in focusing solely on parents as socialization agents, arguing that just as children react to parents, many parent behaviors are also reactive, arising in response to child attributes and behaviors

 55-57 Parents are assumed to play an important role in successful youth development by fostering a sense of belongingness and meaningfulness, and by promoting the development of self-regulation skills connected to competence  Due to the relevance of the aforementioned questions for the strong personality of the children, should it be noted that the benefits of volunteering should be considered a complement, an improvement, to that essential function of parents?

R. We incorporated this thought in the paragraph on page 12.

469  the goal of competent parenting is precisely the encouragement of this sort of disengagement and self-reliance It is not clear that these two characteristics are beneficial for adolescents

R. We have clarified the meaning of the statement within the same paragraph on page 12.

Reviewer 3 Report

see file

Author Response

Our response is in the document attached

Round 2

Reviewer 3 Report

The authors have sufficiently addressed the concerns I had in the first draft. 

Author Response

Thank you for your positive evaluation and constructive suggestions that led to this version of the manuscript.